# Accuracy and reliability of focused echocardiography in patients with Chagas disease from endemic areas: SaMi-Trop cohort study

Isabella Morais Martins Barros[1], Marcio Vinicius L. Barros[1], Larissa Natany Almeida Martins[2], Antonio Luiz P. Ribeiro[1,3], Raul Silva Simões de Camargo[3], Claudia Di Lorenzo Oliveira[4], Ariela Mota Ferreira[5], Lea Campos de Oliveira[6], Ana Luiza Bierrenbach[7], Desireé Sant´Ana Haikal[5], Ester Cerdeira Sabino[8], Clareci S. Cardoso[4], Maria Carmo Pereira Nunes[1,3]*

1 Postgraduate Course of Infectious Diseases and Tropical Medicine, School of Medicine, Universidade Federal de Minas Gerais, Belo Horizonte, MG, Brazil, 2 Department of Statistics, Instituto de Ciências Exatas, Universidade Federal de Minas Gerais. Brazil, 3 Hospital das Clínicas and Faculdade de Medicina, Universidade Federal de Minas Gerais, Belo Horizonte, Brazil, 4 Federal University of São João del-Rei, Divinópolis, Brazil, 5 Health Science Program, Universidade Estadual de Montes Claros, Montes Claros, Brazil, 6 Laboratório de Investigação Médica (LIM03), Hospital das Clinicas da Faculdade de Medicina da Universidade de São Paulo, São Paulo, Brazil, 7 Research and Education Institute–Hospital Sírio-Libanês, São Paulo, Brazil, 8 Instituto de Medicina Tropical e Departamento de Moléstias Infecciosas e Parasitarias da Faculdade de Medicina da Universidade de São Paulo, São Paulo, Brazil

* mcarmo@waymail.com.br

## Abstract

### Background

Chagas disease remains a major cause of cardiovascular death in endemic areas. Focused echocardiography (FoCUS) is a point-of-care means of assessing cardiac function which can be useful for the diagnosis of cardiac involvement.

### Objective

This study aims evaluating the characteristics of validity and reliability of FoCUS applied on Chagas disease patients.

### Methods

Patients with Chagas disease coming from an endemic area were selected from a large cohort (SaMi-Trop). A simplified echocardiogram with only three images was extracted from the conventional echocardiogram performed in this cohort. The images were evaluated by an observer who was blinded to the clinical and echocardiographic data, to determine the accuracy and reliability of FoCUS for cardiac assessment. The analysis constituted of 5 pre-specified variables, dichotomized in absence or presence: left ventricular (LV) size and systolic function, right ventricular (RV) size and systolic function, and LV aneurysm.

**Data Availability Statement:** All relevant data are within the paper and its Supporting Information files.

**Funding:** This study was supported by the National Institutes of Health in the form of funds to ECS [Federal Award Identifier Number (FAIN): U19AI098461]. The National Council for Scientific and Technological Development (CNPq) provided scholarships for MCPN and ECS, and the CAPES Foundation, Ministry of Brazil provided a postdoctoral scholarship for CSC [BEX 2477/15-7]. The CNPq [310679/2016-8; 465518/2014-1] and the Minas Gerais Research Foundation (FAPEMIG) [PPM-00428-17; RED-00081-16] provided partial support in the form of funds to ALPR.

**Competing interests:** The authors have declared that no competing interests exist.

## Results

We included 725 patients with a mean age of 63.4 ± 12.3 years, 483 (67%) female. Abnormal electrocardiogram was observed in 81.5% of the patients. Left and right ventricular dysfunctions were found in 103 (14%) and 49 (7%) of the patients, respectively. Sensitivity, specificity, positive predictive value and negative predictive value were 84%, 94%, 70% and 97% for LV enlargement and 81%, 93%, 68% and 97% for LV systolic dysfunction, respectively, and 46%, 99%, 60% and 98% for RV dilatation, and 37%, 100%, 100% and 96% for RV dysfunction, respectively. Inter and intraobserver agreement were 61% and 87% for LV enlargement and 63% and 92% for LV dysfunction, respectively, and 50% and 49% for RV size and 46% and 79% for RV dysfunction, respectively. LV apical aneurysm was found in 45 patients (6.2%) with the lowest sensitivity of FoCUS study (11%; 95% CI 2–28%).

## Conclusions

FoCUS showed satisfactory values of validity and reliability for assessment of cardiac chambers in patients with Chagas disease, except for apical aneurysm. This tool can identify heart disease with potential impact on patient management in the limited-resource setting.

## Introduction

Chagas disease (ChD) remains a serious public health problem in Latina America, affecting 6 million of people [1, 2]. Chagas cardiomyopathy is the most severe manifestation of ChD, which is characterized by ventricular enlargement with impairment of segmental or global systolic function, generally associated with typical electrocardiographic (ECG) abnormalities [3].

Echocardiography is a well-established method in the evaluation of patients with ChD. The quantification of myocardial involvement is currently one of the main method indications, providing essential data for therapeutic management and prognostic stratification [4]. However, the scarcity of resources and qualified professionals, especially in remote areas, hinders the proper evaluation of these patients. Therefore, strategies need to be developed to make it more practical and accessible, especially in the setting with limited technological resources.

Focused echocardiography (FoCUS) represents a targeted, standardized echocardiographic examination performed by a physician or properly trained practitioner, using ultrasound as a complement to physical examination to recognize certain signs in specific clinical contexts [5, 6]. Several studies have demonstrated the importance of FoCUS in various cardiac pathologies, including left ventricular dilatation and hypertrophy, left ventricular systolic function assessment, left atrial dilatation, right ventricular morphofunctional assessment, and pericardial effusion [7–10].

While it is widely tested and validated in emergency and intensive care settings, few studies have evaluated the use of this tool in the context of primary health care for screening in endemic diseases in resource-poor areas [11, 12]. So far there are no studies using FoCUS in the setting of ChD, especially in remote areas. Given this context, the aim of this study was to evaluate the potential of a focused echocardiography protocol in patients with ChD living in remote poor areas to identify features related to cardiac involvement.

## Methods

### Study population

The patients selected in this study come from a large ongoing cohort, resulting from a partnership between researchers from the states of Minas Gerais and São Paulo, through a project

called São Paulo—Minas Gerais Tropical Medicine Research Center (SaMi-Trop). This project has the purpose of developing and conducting research projects on neglected diseases in Brazil, focusing mainly on Chagas disease. This cohort, composed of patients with chronic Chagas cardiomyopathy was established using patients under the care of the Telehealth Network of Minas Gerais Network, a programme designed to support for primary care in the state of Minas Gerais, Brazil, based on ECG results from 2011–2012. Using this database, we selected 21 municipalities within a limited region in the northern part of the State of Minas Gerais where the prevalence of patients with Chagas cardiomyopathy was expected to be high. All eligible participants tested for *T. cruzi* antibodies using Immunofluorescence and Hemoglutination for *T. cruzi*. The final cohort consists of adults patients confirmed as seropositive [13].

Phase 1 of the cohort occurred between June 2013 and August 2014, with ECG record and blood collection–C-reactive protein (PCR) and N-terminal pro-brain natriuretic peptide (NT-ProBNP) in 2,157 participants. The echocardiogram was performed in phase 2 by a healthcare professional trained for image acquisition and took place between June 2015 and September 2016, totaling 1,713 participants.

Standard transthoracic echocardiography (TTE) were performed at the primary care units of reference for each patient by a healthcare professional, using the Vivid Q GE® portable device with HD image storage and specific software analysis (Echopac; GE Healthcare, Milwaukee, WI). Standardized parasternal long-axis and short-axis views were obtained, as well as apical views in two, three and four chambers. Assessment of cardiac chambers, valves and diastolic function were performed as recommended [14] and included in the cohort database.

Subsequently, the stored images were analyzed in the SaMi-Trop core lab by experienced echocardiographers. For the present study, a subset of patients was randomly selected for assessment of the cardiac chambers using focused echocardiography.

### Ethics statement

The study was approved by the Committee for Ethics in Research of the School of Medicine of the University of São Paulo, number 179.685/2012. All participants were adults ($>$ 18 years old) and signed, in writing and in person, the informed consent form to participate in the study.

### Sample size calculation

For this study two sample size calculations were performed. The first calculation was obtained to determine the agreement between the focused echocardiogram and full echocardiogram, aiming at an accuracy of 85%. The sample size was estimated in 195 patients. The second sample was calculated to assess inter-observer variability, using the Kappa (K) concordance test, also known as the Kappa coefficient. We set the Kappa coefficient of 0.85, thus the sample size was estimated in 730 patients. For intra-observer variability, 20% of the sample was used.

Only participants from the SaMi-Trop cohort were included in this study and patients with inadequate image quality were excluded (3%).

### Focused echocardiography

From the full conventional echocardiographic study, an independent observer selected only three images. The three images were in two-dimensional mode, in parasternal long-axis, and apical two- and four-chambers views (Fig 1). The selected images were stored on a hard disk (external HD) and evaluated through specific software (Echopac; GE Healthcare, Milwaukee, WI) blinded to the patient data and the results of standard echocardiographic study.

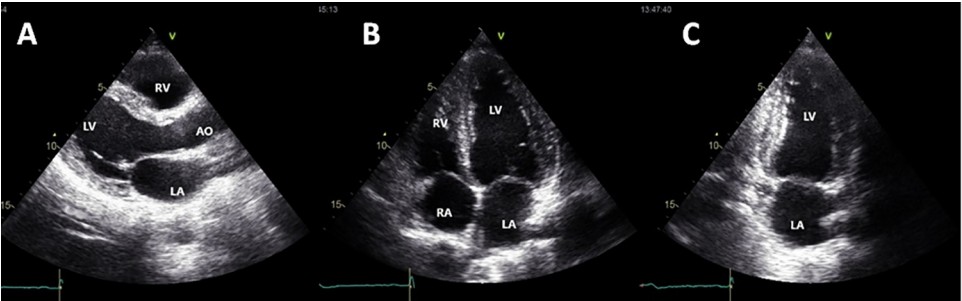

**Fig 1. Echocardiographic views selected for the focused echocardiogram.** A: Parasternal long-axis view; B: Apical four-chamber view; C: Apical two-chamber view. LA: left atrium; RA: right atrium; LV: left ventricle; RV: right ventricle.

The exams were analyzed only by eyeball assessment, without any measurements. The analysis constituted in the five prespecified variables (Fig 2): left ventricular size (0 = normal, 1 = dilated), LV systolic function (0 = normal, 1 = reduced), right ventricular size (0 = normal, 1 = dilated); RV systolic function (0 = normal, 1 = reduced), left ventricular aneurysm (0 = absence, 1 = presence).

## Statistical analysis

Categorical variables, expressed as numbers and percentages, were compared using chi-squared testing, whereas continuous data, expressed as mean ± SD, were compared using Student's unpaired or the Mann-Whitney U test, as appropriate. Correlation analysis were assessed by the Pearson's or Spearman correlation coefficient, as appropriate.

The sensitivity, specificity, positive predictive value (PPV), and negative predictive value (NPV) with 95% confidence interval (CI) of each of the five echocardiographic variables from the focused echo were calculated, considering the results of SaMi-Trop reading as a reference. An agreement between the imaging techniques was evaluated by the weighted Kappa statistic (VassarStats, &Richard Lowry 1998–2011).

Fleiss free-marginal Kappa was used to determine intra and inter observer agreement of the 2 readers given that reviewers were blinded to clinical data. The following standards for strength of agreement for the kappa coefficient were used: poor (0.01 to 0.20); slight (0.21 to 0.40); fair (0.41 to 0.60); moderate (0.61 to 0.80); and excellent (0.81 to 1.00).

Statistical analysis was performed using the Statistical Package for Social Sciences for Windows, version 22.0 (SPSS Inc., Chicago, Illinois) and R for Statistical Computing version 2.15.1 (R Foundation, Vienna, Austria).

## Results

### Patient characteristics

The mean age was 63.4 ± 12.3 years, and 483 were women (67%). The majority were in NYHA functional class I and II at the time of recruitment into the study. The baseline characteristics of the overall population are summarized in Table 1. Previous treatment with benznidazole was reported by 174 patients (24%).

Regarding ECG findings, 134 patients (18.5%) displayed normal ECG. Right bundle-branch block (RBBB) isolated or associated with left anterior fascicular block (LAFB) was the ECG abnormality found in 224 patients (31%), and atrial fibrillation in 32 patients (4.4%) at enrollment (Table 1).

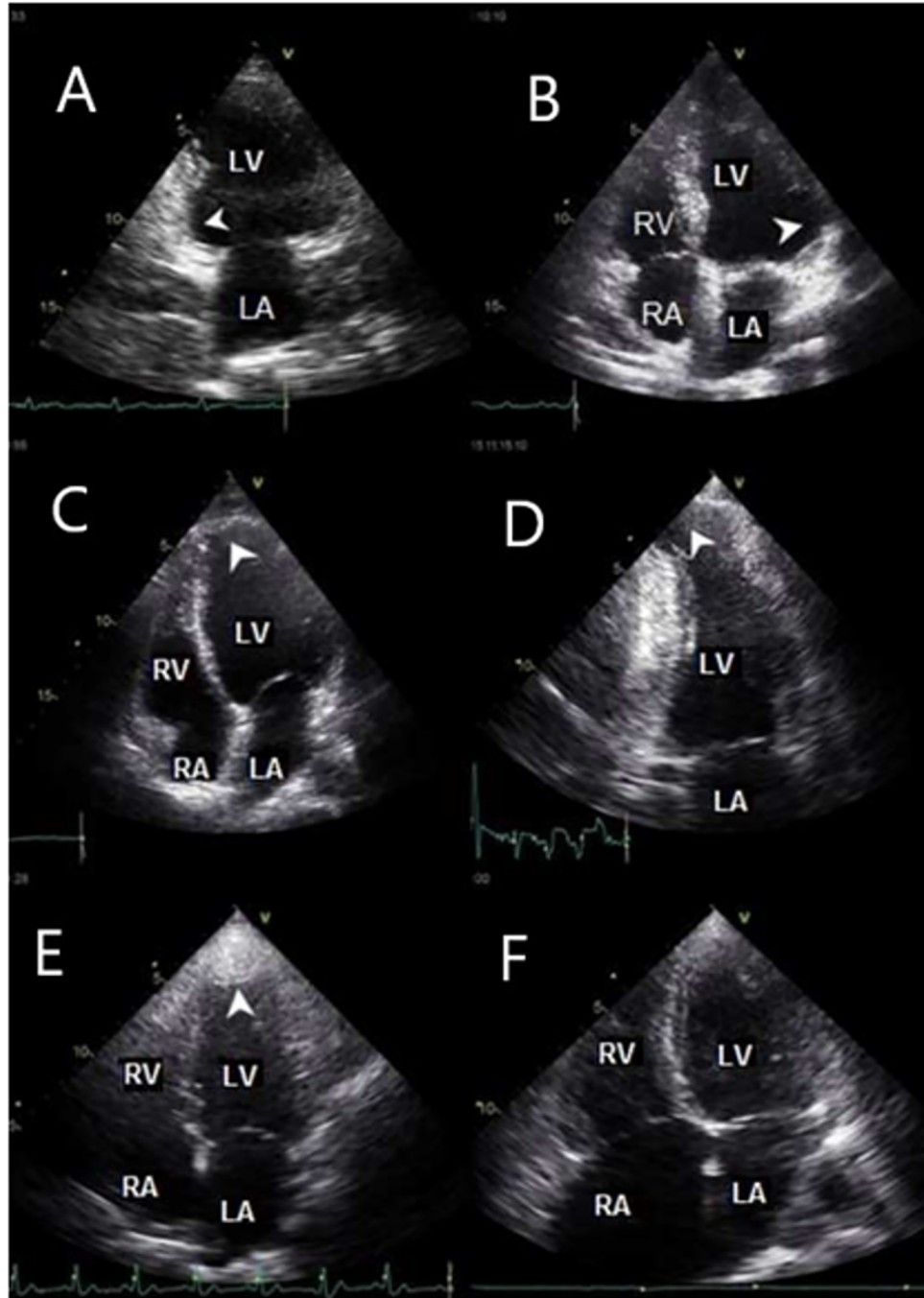

**Fig 2. Echocardiographic studies of Chagas disease patients.** A: akinesia of inferior wall, basal segment; B: akinesia of lateral basal wall, basal segment; C and D: LV apical aneurysm; E: LV apical thrombus; F: right-sided chambers dilation. LA: left atrium; RA: right atrium; LV: left ventricle; RV: right ventricle.

In the overall patient population, 103 patients (14%) presented with LV systolic dysfunction. The echocardiographic parameters are summarized in Table 2. Patients who had normal or minor ECG abnormalities, LV ejection fraction was within the normal limits, whereas those who presented major ECG changes, including frequent ventricular and ectopic beats, atrial

**Table 1. Characteristics of the study population (n = 725).**

| Variables* | | Value |
|---|---|---|
| Age (years) | | 63.4 ± 12.3 |
| Male gender (%) | | 242 (33) |
| NYHA Functional class† | I/II | 459 (63) |
| | III/IV | 252 (35) |
| Syncope‡ | | 78 (11) |
| Palpitations | | 264 (36) |
| Dyslipidemia | | 151 (21) |
| Diabetes mellitus | | 90 (12) |
| Arterial hypertension | | 253 (35) |
| Chronic kidney disease | | 66 (9) |
| Megaesophagus§ | | 77 (11) |
| Megacolon§ | | 46 (6) |
| NT-ProBNP (pg/mL) | | 144 [64/349] |
| **Medications** | | |
| Benznidazole treatment‖ | | 174 (24) |
| ACE inhibitors/ Angiotensin receptor blockers | | 420 (58) |
| Beta blockers (carvedilol) | | 138 (19) |
| Amiodarone | | 167 (23) |
| **ECG findings** | | |
| Heart rate (bpm) | | 66 ± 13.3 |
| Atrial fibrillation | | 32 (4.4) |
| QTc interval (ms) | | 433.4 ± 44.5 |
| PR interval (ms) | | 164.5 ± 35.8 |
| QRS duration (ms) | | 116.6 ± 28.2 |
| Pacemaker | | 22 (3) |
| RBBB¶ | | 224 (31) |
| LBBB | | 24 (3.3) |
| Ventricular ectopic beats | | 15 (2.1) |
| Low QRS voltage | | 40 (5.5) |
| ST-T abnormalities | | 104 (14.3) |

*Data are expressed as the mean value ± SD, median (interquartile range), or absolute numbers (percentage)

†Functional class was not assessed in 14 patients (2%)

‡reported by the patients

§ radiological exams of the gastrointestinal tract reported by the patients.

‖Previous treatment with benznidazole informed by the patients

¶Isolated or associated with left anterior fascicular block (LAFB)

ACE = Angiotensin-converting enzyme inhibitors; LAFB = Left Anterior Fascicular Block, LBBB = Left bundle branch block; NT-ProBNP = N- terminal pro-brain natriuretic peptide; RBBB = Right Bundle Branch Block.

fibrillation, and major isolated ST-T abnormalities had more frequent LV systolic function impairment.

Table 3 shows ECG findings according to the severity of LV systolic dysfunction. Normal ECG was observed in only 4 patients with ejection fraction below 50%, all of them with mild LV dysfunction.

**Table 2. Echocardiographic parameters of the study population.**

| Variables* | Value |
|---|---|
| LV end-diastolic diameter (mm) | 49.1 ± 6.6 |
| LV end-systolic diameter (mm) | 33.1 ± 7.7 |
| LV ejection fraction (%) | 59.7 ± 10.6 |
| LA antero-superior diameter (mm) | 37.4 ± 5.4 |
| LA volume (mL) | 44.1 ± 15.9 |
| E (cm/s) | 67.8 ± 19.2 |
| A (cm/s) | 74.6 ± 20.2 |
| E/A ratio | 0.85 ± 0.4 |
| Deceleration time (ms) | 238.1 ± 54.5 |
| e' septal (cm/s) | 7.2 ± 2.4 |
| E/e' | 10.1 ± 3.9 |
| LV regional wall motion abnormalities | 73 (10) |
| RV diameter† (mm) | 24.3 ± 4.7 |
| LV apical aneurysm | 45 (6.2) |
| RV systolic velocity—S (cm/s) | 12.7 ± 2.8 |
| RV systolic velocity <9.5 (cm/s) | 39 (7.1) |
| RV systolic dysfunction‡ | 49 (6.8) |
| Moderate-severe mitral regurgitation | 103 (14.2) |
| Moderate-severe tricuspid regurgitation | 72 (10) |

*Data are expressed as the mean value ± SD, median (interquartile range), or absolute numbers (percentage).

†Linear dimension measured from the anterior RV wall to the interventricular septal-aortic junction (in parasternal long-axis view).

‡RV systolic dysfunction by qualitative assessment using different two-dimensional views, range from mild to severe dysfunction.

Abbreviations: LA = left atrium; LV = left ventricle; RA = right ventricle.

## FoCUS echocardiography accuracy

Table 4 lists the sensitivity, specificity, negative and positive predictive values, and accuracy for the parameters analyzed between FoCUS and standard echocardiography. The study shows

**Table 3. Electrocardiographic findings according to the severity of left ventricular systolic dysfunction.**

| Values of left ventricular ejection fraction* | ECG data | Number (%) |
|---|---|---|
| Left ventricular ejection fraction <50% and ≥40% (n = 56) | Overall RBBB | 19 (34) |
| | Isolated RBBB | 8 (14) |
| | RBBB plus LAFB | 11 (20) |
| | Normal ECG | 4 (7) |
| Left ventricular ejection fraction <40% and ≥30% (n = 30) | Overall RBBB | 12 (40) |
| | Isolated RBBB | 6 (20) |
| | RBBB plus LAFB | 6 (20) |
| | Normal ECG | 0 |
| Left ventricular ejection fraction <30% (n = 17) | Overall RBBB | 2 (12) |
| | Isolated RBBB | 1 (6) |
| | RBBB plus LAFB | 1 (6) |
| | Normal ECG | 0 |

Left ventricular ejection fraction <50% was found in 103 patients (14%) and ≥50% in the remaining patients.

LVEF = left ventricular ejection fraction.

**Table 4. Accuracy of FoCUS versus Standard Echocardiography in patients with Chagas disease from the Sami-Trop cohort.**

| Variables | Sn | Sp | PPV | NPV | Ac |
|---|---|---|---|---|---|
| LV dilatation | 0.84 (0.75–0.90) | 0.94 (0.91–0.95) | 0.70 | 0.97 | 0.92 |
| LV dysfunction | 0.81 (0.71–0.88) | 0.94 (0.91–0.95) | 0.68 | 0.97 | 0.92 |
| RV dilatation | 0.46 (0.20–0.75) | 0.99 (0.97–0.99) | 0.60 | 0.98 | 0.98 |
| RV dysfunction | 0.37 (0.21–0.56) | 1.0 (0.99–1.0) | 1.0 | 0.95 | 0.96 |
| LV apical aneurysm | 0.11 (0.02–0.28) | 1.0 (0.99–1.0) | 1.0 | 0.94 | 0.94 |

LV = left ventricle; RV = right ventricle; Sn = sensitivity; Sp = specificity; NPV = negative predictive value; PPV = positive predictive value; Ac = accuracy.

excellent accuracy in the evaluation of LV and RV dimension and function using FoCUS, with the highest values obtained from LV study.

The evaluation of LV dilatation showed the best sensitivity (0.84; 95% CI 0.75–0.90) among the variables studied, with false negative readings ranged from 10 to 25%. LV systolic function assessment demonstrated excellent results (sensitivity = 0.81; 95% CI 0.71–0.88), with false positive readings ranged from 5% to 9% only. The presence of apical aneurysm showed the poorest sensitivity with FoCUS study (0.11; 95% CI 0.02–0.28). RV size and systolic function by FoCUS found poor sensitivity and high specificity comparing to standard echocardiography, with excellent accuracy.

**Observer agreement.** Intraobserver agreement was almost perfect in the evaluation of LV size ($\kappa$ = 0.87; 95% CI 0.77–0.97) and function ($\kappa$ = 0.92; 95% CI 0.83–0.99), with substantial agreement in the apical aneurysm ($\kappa$ = 0.66; 95% CI 0.05–1.0) and RV size ($\kappa$ = 0.79; 95% CI 0.40–1.0) assessment. RV function showed moderate intraobserver agreement ($\kappa$ = 0.49; 95% CI 0.11–1.0). The evaluation of interobserver agreement (Table 5) showed substantial agreement from LV size ($\kappa$ = 0.61; 95% CI 0.51–0.74), LV function ($\kappa$ = 0.63; 95% CI 0.53–0.74) and apical aneurysm ($\kappa$ = 0.66; 95% CI 0.05–1.0) and moderate agreement from RV size ($\kappa$ = 0.50; 95% CI 0.31–0.69) and RV function ($\kappa$ = 0.46; 95% CI 0.26–0.66).

## Discussion

Our study evaluated for the first time the role of FoCUS in a large cohort of ChD from remote areas. The main results can be summarized as follows: (i) there was an excellent accuracy in

**Table 5. Intraobserver agreement of FoCUS in patients with Chagas disease from the Sami-Trop cohort.**

| Variables | Kappa | 95% CI |
|---|---|---|
| **Intraobserver agreement** | | |
| LV dilatation | 0.87 | 0.77–0.97 |
| LV dysfunction | 0.92 | 0.83–0.99 |
| RV dilatation | 0.49 | 0.11–1.0 |
| RV dysfunction | 0.79 | 0.40–1.0 |
| LV apical aneurysm | 0.66 | 0.05–1.0 |
| **Interobserver agreement** | | |
| LV dilatation | 0.61 | 0.50–0.71 |
| LV dysfunction | 0.63 | 0.53–0.73 |
| RV dilatation | 0.50 | 0.31–0.69 |
| RV dysfunction | 0.46 | 0.26–0.66 |
| LV apical aneurysm | 0.66 | 0.05–1.0 |

LV = left ventricle; RV = right ventricle; CI = confidence interval.

the evaluation of LV and RV dimension and function; (ii) there was a moderate to substantial intra and interobserver agreement in the FoCUS assessment.

Chagas disease is a potentially lethal condition, but its severity varies widely. Chagas cardiomyopathy is the most serious clinical manifestation of the disease, which is defined by the presence of typical ECG abnormalities in the patients who have positive serological testes against *Trypanosoma cruzi* [3]. Therefore, identification of heart involvement is essential for accurate risk stratification [15]. Several risk scores have been proposed and developed. However, current risk scores rely on the availability of several diagnostic tests, including echocardiographic examination [15–17]. These methods are not readily available in the rural endemic areas and a lack of a health service structure, mainly in remote areas, along with the low levels of awareness among healthcare providers, which leads to cases of chronic Chagas cardiomyopathy be under-recognized.

The SaMi-Trop is one of the largest multicentre cohort study of ChD conducted in the world [13]. The large number of patients included in this investigation is outstanding, especially in a rural and dispersed area. Strategies as focused echocardiography performed by non-physicians and using telehealth in reference centers could facilitate access to more accurate diagnosis, enhancing health care quality in under-serviced communities.

The assessment of the LV function plays a fundamental role in the evaluation of diagnosis and prognosis of chronic ChD. Echocardiography is the noninvasive method of choice in clinical practice for the detection of abnormalities in LV size and function [4]. Patients with ChD may present with predominantly hypokinetic, dilated with diminished LV ejection fraction (LVEF), or biventricular dilatation [18]. Some studies have shown the association between LV diameter and prognosis in patients with Chagas cardiomyopathy. Viotti et al. in a cohort of 856 patients with an 8-year follow-up demonstrated that LV end-systolic diameter was an independent predictor of disease progression and cardiovascular mortality [19]. LV end-systolic diameter was also an independent predictor of mortality or heart transplantation in a study with 231 patients and 19-month follow-up [20]. Issa et al. evaluating Chagas disease patients with irreversible chronic heart failure observed that LV end-diastolic diameter was an independent and significant predictor of mortality or heart transplantation [21].

LV global dysfunction, usually expressed by low LVEF, is the most important predictor of death in Chagas disease [15, 22]. Several studies have demonstrated the prognostic value of systolic dysfunction in patients with ChD. In a systematic review, echocardiographic or cineventriculographic evidence of contractility reduction, expressed qualitatively or quantitatively, was strongly associated with an increased risk of mortality in most studies evaluated [23].

## Relevance of the echocardiogram in the evaluation of patients with Chagas disease

In overall population with Chagas disease, echocardiographic evaluation should be performed to assess LV function, regardless of ECG findings. Although the lack of ECG abnormalities mostly rules out significant cardiomyopathy, it is reasonable to perform at least a single echocardiogram as a baseline evaluation on every patient with positive serology for Chagas disease [3]. Individuals with normal ECG may display subtle regional wall motion abnormalities or ventricular aneurysm, which may impact on patient management and follow-up frequency [4]. Echocardiogram should be repeated during follow-up if the ECG becomes abnormal to establish disease progression [31]. In rural areas with very limited resources, a focused echocardiogram could be used as a screening in asymptomatic individuals, which may provide incremental information to ECG.

More recent guidelines recommend that it is reasonable to obtain at least one echocardiogram for patients diagnosed with Chagas disease [3, 4, 31]. This study establishes a baseline for later comparison and can detect the small percentage of patients with subclinical abnormalities despite a normal 12-lead ECG. Echocardiogram should be repeated every 3–5 years in the patients with preserved left ventricular ejection fraction and more often in those who have reduced ejection fraction at the diagnosis or when clinical status change with worsening heart failure or embolic events. The recommendations for echocardiogram in patients with Chagas disease according to the current guidelines are shown in Table 6.

## Screening for cardiac dysfunction in Chagas disease: role of FoCUS

Several studies have demonstrated the accuracy of FoCUS in the study of LV, with sensitivity and specificity for size determination varying from 73–100% and 64–93%, respectively and with accuracy superior to 90% in the analysis of LV function [27–29]. Our study shows an excellent accuracy and observer concordance in the evaluation of LV size and function. These findings show the potential for a widespread use of FoCUS as a pivotal protocol in the evaluation of ChD patients in endemic areas. A proposed approach based on LV systolic function assessed by FoCUS is shown in Fig 3. Patients who presented LV systolic dysfunction should be referred for a comprehensive echocardiographic evaluation, regardless of ECG findings.

LV apical aneurysm, a specific and important characteristic in ChD have a prevalence 6.2% in our cohort, slightly lower than previous studies [18, 30]. However, we obtained poor sensitivity for the evaluation of apical aneurysms. These results could be related to the complex characteristic of apical aneurysm in ChD, generally being necessary several maneuvers for the appropriate assessment of apical region during echocardiographic study. Apical aneurysm may be missed if only conventional apical views are acquired. In order to identify aneurysms, a careful examination requires not only standard views but also angulated apical views. Frequently, a modified four- and two-chambers views aiming posteriorly may be necessary to

**Table 6. Recommendations for echocardiogram in Chagas disease.**

| Guidelines (author/year) | Main indications |
|---|---|
| I Latin American Guidelines for the Diagnosis and Treatment of Chagas' Heart Disease (Andrade J.P/2011) [24] | Additional diagnostic and prognostic assessment of patients with Chagas heart disease with abnormal ECG |
| Brazilian Consensus on Chagas Disease (Dias J.C.P/2015) [25] | Abnormal ECG to classify myocardial damage into stages |
| Multimodality imaging evaluation of Chagas disease: an expert consensus of Brazilian Cardiovascular Imaging Department (DIC) and the European Association of Cardiovascular Imaging (EACVI) (Nunes M.C.P/2017) [31] | It is reasonable to perform an echocardiogram on every patient at the diagnosis of Chagas disease, and it should be repeated during follow-up if the ECG becomes abnormal |
| Recommendations for Multimodality Cardiac Imaging in Patients with Chagas Disease: A Report from the American Society of Echocardiography in Collaboration With the InterAmerican Association of Echocardiography (ECOSIAC) and the Cardiovascular Imaging Department of the Brazilian Society of Cardiology (DIC-SBC) (Acquatella H/2018) [4] | It is reasonable to perform at least a single echocardiographic examination (baseline evaluation) on every patient with positive serology for Chagas disease and repeat during follow-up if the ECG findings become abnormal to document disease progression |
| Chagas Cardiomyopathy: An Update of Current Clinical Knowledge and Management A Scientific Statement From the American Heart Association (Nunes M.C.P/2018) [3] | It is reasonable to obtain at least 1 echocardiogram for patients diagnosed during the indeterminate stage of Chagas disease. |
| Chagas Disease Consensus–Argentine Society of Cardiology (Benassi MD/2019) [26] | It is indicated in the initial assessment of Chagas disease and when new symptoms or ECG changes. |

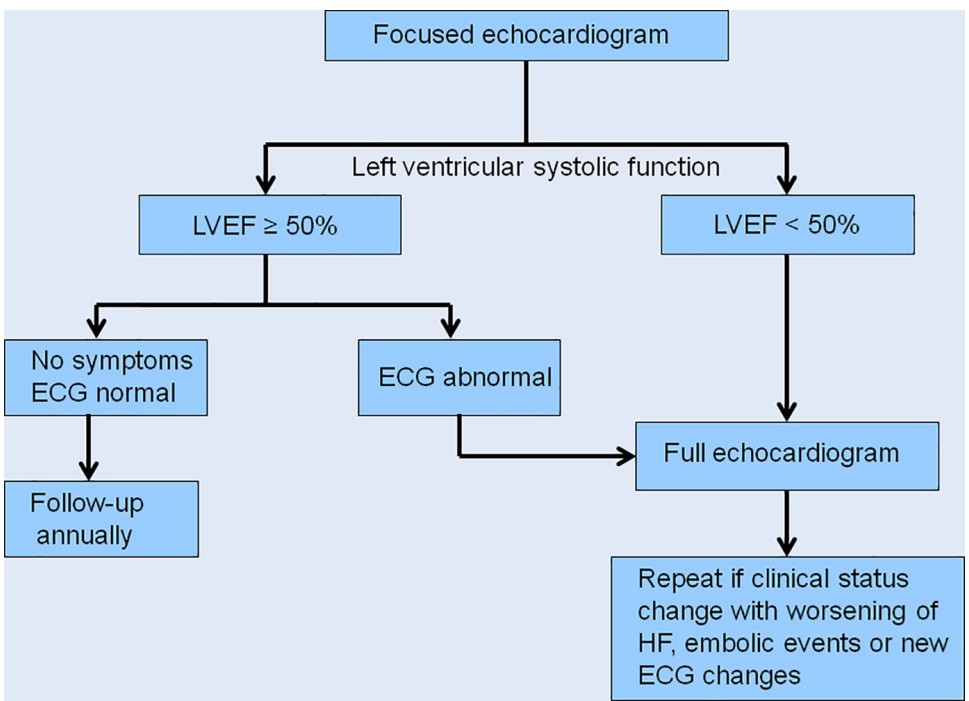

**Fig 3. Management of patients with Chagas disease based on FoCUS results.** A proposed approach based on left ventricular systolic function assessed by focused echocardiography.

detect apical aneurysms and thrombus [31]. Therefore, the low sensibility to detect aneurysm in FoCUS is expected. For this specific abnormality, a comprehensive echocardiographic examination is warrant to avoid misdiagnosis of both aneurysm and thrombus, which is often associated with aneurysms.

Right ventricular involvement is common in ChD and it is clear that there is a progression of RV dysfunction among patients with the various forms of ChD [32]. Nevertheless, the echocardiographic study of RV is challenging because of the anatomical and functional complexity of this chamber and the visual assessment of RV size and function could be difficult [14]. Previous studies have investigated the reliability of RV function assessment using "eyeballing" alone showing inconsistent results [33, 34]. In a recent study comparing visual detection of reduced RV function using a RV-focused 4-chamber view comparing with cardiac magnetic resonance (derived RV ejection fraction <50%), sensitivity was 97.1%, 96,8%, 96.5%, and 95.8% and specificity was 55.7%, 52.8%, 54.6%, and 42.5% for the expert, advanced, intermediate, and beginner groups, respectively. For determination of the correct degree of RV dysfunction, even experienced examiners assigned a diagnosis that was discordant with CMR in > 40% of cases, suggesting that visual assessment should be combined with measurement of other parameters of RV function [35]. Ling et al in a study comparing RV evaluation by echocardiography and CMR showed that visual estimation of RV size and function was inaccurate and had wide interobserver variability [36]. Our results demonstrated an excellent accuracy for RV size evaluation and a poor sensitivity and excellent specificity for RV function showing the challenging assessment of this chamber by FoCUS, in agreement with previous studies.

Observer agreement showed good results for visual assessment of LV size and function using FoCUS as previously demonstrated by several studies [27, 29]. However, different study designs and population characteristic among the studies limits an appropriated comparison. Interobserver variability for RV assessment showed only moderate agreement in our study

reinforcing the challenge for this chamber evaluation by FoCUS using only visual assessment [35, 36].

## Study limitations

This study has several limitations. Echocardiographic studies were performed by non-physicians and could affect the final echocardiographic interpretation. Training for these health professionals consisted of practical training supervised by a cardiologist (MN) at the University's Echocardiography Laboratory (12 weeks). The interpretation of echocardiograms was limited to cardiologists with extensive experience in echocardiography that probably increased reliability and cannot necessarily be reproduced in other settings.

Additionally, the size of the left and right atria were not evaluated, which is important in the diagnosis and management of patients with Chagas cardiomyopathy. Moreover RV focused apical 4-chamber view was not obtained, which would likely improve the sensitivity of both size and systolic RV function by means of eyeball evaluation.

## Conclusions

FoCUS presented satisfactory accuracy and agreement in the morphofunctional assessment of cardiac chambers compared to conventional echocardiography in patients with chronic Chagas disease from endemic areas. This tool may be useful for patient management, especially in the limited-resource setting with difficult access of health care.

## Author Contributions

**Conceptualization:** Isabella Morais Martins Barros, Marcio Vinicius L. Barros, Maria Carmo Pereira Nunes.

**Data curation:** Raul Silva Simões de Camargo.

**Formal analysis:** Isabella Morais Martins Barros, Larissa Natany Almeida Martins, Ana Luiza Bierrenbach, Clareci S. Cardoso, Maria Carmo Pereira Nunes.

**Funding acquisition:** Ester Cerdeira Sabino.

**Methodology:** Antonio Luiz P. Ribeiro, Ariela Mota Ferreira, Lea Campos de Oliveira, Desireé Sant´Ana Haikal, Ester Cerdeira Sabino, Maria Carmo Pereira Nunes.

**Project administration:** Claudia Di Lorenzo Oliveira, Ester Cerdeira Sabino.

**Software:** Raul Silva Simões de Camargo.

**Supervision:** Ariela Mota Ferreira, Lea Campos de Oliveira, Ana Luiza Bierrenbach, Ester Cerdeira Sabino, Maria Carmo Pereira Nunes.

**Writing – original draft:** Isabella Morais Martins Barros.

**Writing – review & editing:** Marcio Vinicius L. Barros, Antonio Luiz P. Ribeiro, Claudia Di Lorenzo Oliveira, Clareci S. Cardoso, Maria Carmo Pereira Nunes.

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
