## [Decision Letter · Decision Letter 0]

16 Jun 2021

PONE-D-20-35550

Accuracy and reliability of focused echocardiography in patients with Chagas disease from endemic areas: SaMi-Trop Cohort Study

PLOS ONE

Dear Dr. Pereira Nunes,

Thank you for submitting your manuscript to PLOS ONE. After careful consideration, we feel that it has merit but does not fully meet PLOS ONE’s publication criteria as it currently stands. Therefore, we invite you to submit a revised version of the manuscript that addresses the points raised during the review process.

We look forward to receiving your revised manuscript.

Kind regards,

Daniel A. Morris, M.D

Academic Editor

PLOS ONE

Journal Requirements

- https://escholarship.org/content/qt2fp014zq/qt2fp014zq.pdf?t=qadwdi

https://academic.oup.com/ehjcimaging/article/19/4/459/4222661

https://pubmed.ncbi.nlm.nih.gov/27147390/

In your revision ensure you cite all your sources (including your own works), and quote or rephrase any duplicated text outside the methods section. Further consideration is dependent on these concerns being addressed."

3. Thank you for including your ethics statement: "The SaMi-Trop Cohort was approved by the Institutional Review Board, number 179.685/2012 (National Commission of Ethics in Research, CONEP)."   

4. Please provide additional details regarding participant consent. In the ethics statement in the Methods and online submission information, please ensure that you have specified (1) whether consent was informed and (2) what type you obtained (for instance, written or verbal, and if verbal, how it was documented and witnessed). If your study included minors, state whether you obtained consent from parents or guardians. If the need for consent was waived by the ethics committee, please include this information.

Editor Comments:

I would like to congratulate to the authors for this excellent study addressing the potential usefulness of portable and focused echocardiography in remote areas of South America for screening a forgotten and ignored serious CV disease such as Chagas, which affects mainly to the poor people of South America. I have only some minor comments to increase even more the clinical relevance of this interesting and clinically relevant study:

1) The reviewer has stated very important issues, which should be mandatorily addressed in the revised version of the manuscript.

2) The authors should perform an additional table showing the following findings:

- rate of LV dysfunction defined as LVEF < 50% in all patients, in patients with RBBB, in patients with RBBB + LAFB, in patients with isolated RBBB, and in patients with normal ECG.

- rate of LV dysfunction defined as LVEF < 40% in all patients, in patients with RBBB, in patients with RBBB + LAFB, in patients with isolated RBBB, and in patients with normal ECG.

- rate of LV dysfunction defined as LVEF < 30% in all patients, in patients with RBBB, in patients with RBBB + LAFB, in patients with isolated RBBB, and in patients with normal ECG.

3) The authors should discuss with details and comprehensively the potential clinical relevance for the detection and diagnosis of Chagas cardiomyopathy of focused and portable echocardiography in comparison with an isolated ECG. In other words, why we should perform a TTE in seropositive patients regardless of a normal ECG ?; should we do a TTE in patients with normal ECG ?; should all patients with any change in ECG have a TTE or only in patients with RBBB ?

4) The authors should show in an additional table and discuss with details and comprehensively what did state the Chagas Guidelines of the ASE, of the Brazilian Society of Cardiology, and of the Argentinian Society of Cardiology about the use and indication of TTE (focused or not, portable or not) in seropositive patients with normal or abnormal ECG for the screening or detection of Chagas cardiomyopathy.

5) The authors should show in an additional table and discuss with details and comprehensively what did state the Chagas Guidelines of the ASE, of the Brazilian Society of Cardiology, and of the Argentinian Society of Cardiology about the definition or diagnostic criteria of Chagas cardiomyopathy.

6) The authors should discuss with details and comprehensively how we should procedure to diagnose Chagas cardiomyopathy in a seropositive patient with LVEF < 50% and AF or CAD.

7) The authors should show in a figure and discuss with details and comprehensively how we should manage a seropositive patient with a recent diagnosis of LVEF < 50% by portable or focused TTE.

Reviewers' comments:

Reviewer's Responses to Questions

**Comments to the Author**

1. Is the manuscript technically sound, and do the data support the conclusions?

Reviewer #1: Yes

2. Has the statistical analysis been performed appropriately and rigorously? 

Reviewer #1: Yes

3. Have the authors made all data underlying the findings in their manuscript fully available?

Reviewer #1: Yes

4. Is the manuscript presented in an intelligible fashion and written in standard English?

Reviewer #1: Yes

5. Review Comments to the Author

Reviewer #1: GENERAL COMMENTS:

The subject is of global interest, considering the worldwide spread of the disease due to the crescent human migration. Additionally, an objective and quick approach of the patients regarding the cardiac evaluation is of great importance mainly when considering the less favoured areas, which are numerous in poor countries.

Overall, the text is written in standard English; however, two issues must be observed:

Of note, the punctuation mark “full stop” needs to come before the citation number (e.g: Introduction, 1st paragraph- “Chagas disease (ChD) remains a serious public health problem in Latina America, affecting 6 million of people.1,2”).

Introduction, 1st paragraph: “Chagas cardiomyopathy is the most severe manifestation of ChD, which is characterized by ventricular enlargement with impairment of systolic function segmental or global associated with typical electrocardiographic (ECG) abnormalities3.” I would suggest to rewrite the phrase as: “Chagas cardiomyopathy is the most severe manifestation of ChD, which is characterized by ventricular enlargement with impairment of segmental or global systolic function, generally associated with typical electrocardiographic (ECG) abnormalities.3” It seems to me more correct in terms of English Grammar.

METHODS

Left atrial size, which is a cardinal parameter in terms of prognosis,1, 2 was not included in the eyeball analysis. Likewise, it would be of interest to include the right atrial size in the evaluation. Additionally, 4-chamber view focused on right ventricle was not performed, according to guidelines and cited by the authors in the text (reference 14).

DISCUSSION

P 11, 2nd paragraph: “In order to identify aneurysms, a careful examination requires not only standard views but also angulated apical views. Frequently, a modified four- and two-chambers views aiming posteriorly may be necessary to detect apical aneurysms and thrombus. Additionally, there is a broadly definition of aneurysm depending on criteria used, which may range from a small akinetic area to large with extensive wall thinning, similar to ischemic aneurysms18.Therefore, the low sensibility to detect aneurysm in FoCUS is expected”

The study should have included extra careful examination, using modified apical views aiming at more accurate diagnosis of apical aneurysms. Despite the small ones may not alter the LVEF, 3 they represent an abnormality related to the disease and an attentive observation is needed. Considering the “broad definition of aneurysm depending on the criteria used”, it does not seem to me a proper argument, because one may always describe the alteration; additionally, the stored images were all subsequently analysed by experienced ecocardiographers (p. 5, 2nd paragraph, lines 14 and 15), who can standardise the definition of apical aneurysm.

P 12, 1st paragraph: “Right ventricular involvement is common in ChD and it is clear that there is a progression of RV dysfunction among patients with the various forms of ChD28. Nevertheless, the echocardiographic study of RV is challenging because of the anatomical and functional complexity of this chamber and the visual assessment of RV size and function could be difficult29... Our results demonstrated an excellent accuracy for RV size evaluation and a poor sensitivity...”

I would like to hear from the authors why they did not perform modified apical 4C view focused on RV: this would be easy, no time consuming, and would probably improve the sensitivity of both size and systolic function by means of eyeball evaluation.

FIGURE 2: Letters A, B, C, D, E and F are missing

REFERENCES

1. Hoit BD. Left atrial size and function: role in prognosis. J Am Coll Cardiol.2014;63(6):493-505

2. Saraiva RM, Pacheco NP, Pereira TOJS et al. Left atrial structure and function predictors of new-onset atrial fibrillation in patients with Chagas Disease. J Am Soc Echocardiogr.2020;33(11):1363-1374

3. Nogueira EA, OM Ueti, WR Vieira. The apical ventricular lesion in Chagas heart disease. Sao Paulo Med J. 1995;113(2):785-790

---

## [Author Response · Author response to Decision Letter 0]

26 Aug 2021

Manuscript number: PONE-D-20-35550

Title: Accuracy and reliability of focused echocardiography in patients with Chagas disease from endemic areas: SaMi-Trop Cohort Study

Thank you for considering our manuscript for publication in Plos One. We are re-submitting a revised document in which we have attempted to address the Reviewers’ concerns and the editorial comments. Please see a detailed point-by-point response below.

 Journal Requirements

Both the Editorial Committee and Statistical Reviewer have serious concerns about the data presentation and analysis. However, we are willing to consider a revision if these concerns can be adequately addressed either by changes in the manuscript or acknowledgement in the limitations section.

Response: We have revised the manuscript to meet PLOS ONE style requirements and the changes are highlighted in the revised manuscript.

- https://escholarship.org/content/qt2fp014zq/qt2fp014zq.pdf?t=qadwdi: BMJ Open 2016 May 4;6(5):e011181 - - https://academic.oup.com/ehjcimaging/article/19/4/459/4222661: Eur Heart J Card Imag (2018) 19, 459–460 

- https://pubmed.ncbi.nlm.nih.gov/27147390/ BMJ Open 2016 May 4;6(5):e011181

In your revision ensure you cite all your sources (including your own works), and quote or rephrase any duplicated text outside the methods section. Further consideration is dependent on these concerns being addressed."

Response: 

Two links (https://escholarship.org/content/qt2fp014zq/qt2fp014zq.pdf?t=qadwdi AND https://pubmed.ncbi.nlm.nih.gov/27147390) refer to the same publication on SaMi-Trop cohort profile in which our study is included (Longitudinal study of patients with chronic Chagas cardiomyopathy in Brazil (SaMi-Trop project): a cohort profile. We refer to the description of the study population, included in the Methods section, on page 5, 1st paragraph, with the publication that described the characteristics of this cohort and referenced with number 13.

We reviewed the manuscript and included reference number 31 (from our group) to the manuscript (https://academic.oup.com/ehjcimaging/article/19/4/459/4222661;Multimodality imaging evaluation of Chagas disease: an expert consensus of Brazilian Cardiovascular Imaging Department (DIC) and the European Association of Cardiovascular Imaging (EACVI) European Heart Journal - Cardiovascular Imaging (2018) 19, 459–460) 

Thank you for including your ethics statement: "The SaMi-Trop Cohort was approved by the Institutional Review Board, number 179.685/2012 (National Commission of Ethics in Research, CONEP)." Please amend your current ethics statement to include the full name of the ethics committee/institutional review board(s) that approved your specific study. Once you have amended this/these statement(s) in the Methods section of the manuscript, please add the same text to the "Ethics Statement" field of the submission form (via "Edit Submission").

Response: The present study is a substudy of a large multicentre cohort called SaMi-Trop, which assesses epidemiological, clinical, laboratory, electrocardiographic and echocardiographic data. The study was approved by the Committee for Ethics in Research of the School of Medicine of the University of São Paulo. The full name of the ethics committee/institutional review board that approved the study has been included.

 Please provide additional details regarding participant consent. In the ethics statement in the Methods and online submission information, please ensure that you have specified (1) whether consent was informed and (2) what type you obtained (for instance, written or verbal, and if verbal, how it was documented and witnessed). If your study included minors, state whether you obtained consent from parents or guardians. If the need for consent was waived by the ethics committee, please include this information.

Response: All participants were adults (> 18 years old) who signed, in writing and in person, the informed consent form to participate in the study. This information has been added to the manuscript on page 5, Methods section. 

Editor Comments:

1) The authors should perform an additional table showing the following findings:

- rate of LV dysfunction defined as LVEF < 50% in all patients, in patients with RBBB, in patients with RBBB + LAFB, in patients with isolated RBBB, and in patients with normal ECG.

- rate of LV dysfunction defined as LVEF < 40% in all patients, in patients with RBBB, in patients with RBBB + LAFB, in patients with isolated RBBB, and in patients with normal ECG.

- rate of LV dysfunction defined as LVEF < 30% in all patients, in patients with RBBB, in patients with RBBB + LAFB, in patients with isolated RBBB, and in patients with normal ECG.

Response: A table with these findings has been added to the manuscript in Results section on page 11 (Table 3).

2) The authors should discuss with details and comprehensively the potential clinical relevance for the detection and diagnosis of Chagas cardiomyopathy of focused and portable echocardiography in comparison with an isolated ECG. In other words, why we should perform a TTE in seropositive patients regardless of a normal ECG? Should we do a TTE in patients with normal ECG? Should all patients with any change in ECG have a TTE or only in patients with RBBB?

Response: In overall population with Chagas disease, echocardiographic evaluation should be performed to assess LV function, regardless of ECG findings. Although the lack of ECG abnormalities mostly rules out significant cardiomyopathy, it is reasonable to perform at least a single echocardiogram as a baseline evaluation on every patient with positive serology for Chagas disease. Individuals with normal ECG may display subtle regional wall motion abnormalities or ventricular aneurysm, which may impact on patient management and follow-up frequency. In rural areas with very limited resources, a focused echocardiogram could be used as a screening in asymptomatic individuals, which may provide incremental information to ECG. Echocardiogram should be repeated during follow-up if the ECG becomes abnormal to establish disease progression or when clinical status change with worsening heart failure or embolic events.

A topic regarding the value of the echocardiogram in the evaluation of patients with Chagas disease has been added to the manuscript in the Discussion, on page 14. 

3) The authors should show in an additional table and discuss with details and comprehensively what did state the Chagas Guidelines of the ASE, of the Brazilian Society of Cardiology, and of the Argentinian Society of Cardiology about the use and indication of TTE (focused or not, portable or not) in seropositive patients with normal or abnormal ECG for the screening or detection of Chagas cardiomyopathy.

Response: A Table summarizing the recommendations for echocardiogram in patients with Chagas disease has been added to the manuscript (Table 6). 

4) The authors should show in an additional table and discuss with details and comprehensively what did state the Chagas Guidelines of the ASE, of the Brazilian Society of Cardiology, and of the Argentinian Society of Cardiology about the definition or diagnostic criteria of Chagas cardiomyopathy.

Response: According to the guidelines, Chagas cardiomyopathy is defined by the presence of typical electrocardiographic abnormalities in the patients who have positive serological testes against Trypanosoma cruzi. ECG abnormalities are often the first indicator of cardiac involvement in Chagas disease. The most common ECG abnormalities are right bundle branch block, left anterior fascicular block, ventricular premature beats, ST-T changes, abnormal Q waves, and low voltage of QRS. However, no single ECG finding is pathognomonic for Chagas cardiomyopathy. 

This definition has been added to the manuscript on page 13.

5) The authors should discuss with details and comprehensively how we should procedure to diagnose Chagas cardiomyopathy in a seropositive patient with LVEF < 50% and AF or CAD.

Response: Atrial fibrillation is the most common supraventricular arrhythmia, found in up to 5% of electrocardiographic tracings of patients with chronic Chagas disease. Most typically, atrial fibrillation is seen in patients with dilated cardiomyopathy, often indicating advanced myocardial damage. Therefore, a seropositive patient with LVEF < 50% and atrial fibrillation should be managed following the recommendations for heart failure treatment, including 

anticoagulation therapy.

6) The authors should show in a figure and discuss with details and comprehensively how we should manage a seropositive patient with a recent diagnosis of LVEF < 50% by portable or focused TTE.

Response: A Figure with a proposed approach based on left ventricular systolic function assessed by focused echocardiography has been added to the manuscript (Figure 3).

Reviewer Comments:

1. Overall, the text is written in standard English; however, two issues must be observed: 

1.1. Of note, the punctuation mark "full stop" needs to come before the citation number (e.g: Introduction, 1st paragraph- "Chagas disease (ChD) remains a serious public health problem in Latina America, affecting 6 million of people.1,2").

Response: Thank you for your observation. Regarding sentence punctuation, we followed the PLOS ONE publication style guidelines, with punctuation after the reference written in parentheses being a requirement.

1.2. Introduction, 1st paragraph: "Chagas cardiomyopathy is the most severe manifestation of ChD, which is characterized by ventricular enlargement with impairment of systolic function segmental or global associated with typical electrocardiographic (ECG) abnormalities3." I would suggest to rewrite the phrase as: "Chagas cardiomyopathy is the most severe manifestation of ChD, which is characterized by ventricular enlargement with impairment of segmental or global systolic function, generally associated with typical electrocardiographic (ECG) abnormalities.3" It seems to me more correct in terms of English Grammar.

Response: As suggested, the change was made in the Introduction, 1st paragraph on page 3 of the manuscript.

2. Left atrial size, which is a cardinal parameter in terms of prognosis, was not included in the eyeball analysis. Likewise, it would be of interest to include the right atrial size in the evaluation. Additionally, 4-chamber view focused on right ventricle was not performed, according to guidelines and cited by the authors in the text.

Response: Although the left atrial size measurement is an integral part of the analysis of the diastolic function with prognostic value in Chagas disease, this parameter was not obtained in the current study. Left atrial size depends on body surface, which limits its use in the eyeball analysis by focused echo. 

We acknowledge that this is a limitation of the study and this information has been added to the study limitation, on page 18, as follows:

"Additionally, the size of the left and right atria were not evaluated, which is important in the diagnosis and management of patients with with Chagas cardiomyopathy"

RV focused apical 4-chamber view is highly dependent on probe rotation which can result in an underestimation of RV width. Our main goal in the present study is an initial quantification of RV function by visual assessment, which is the most common used parameter in daily clinical practice. 

Previous studies have been demonstrated the inaccuracy of eyeball RV function evaluation comparing with magnetic resonance imaging. Even among experts, classification of the degree of RV dysfunction is imprecise to differentiate between normal and abnormal RV function. 

This has been explained in the discussion section of the manuscript, on page 15, 1st paragraph.

3. P 11, 2nd paragraph: "In order to identify aneurysms, a careful examination requires not only standard views but also angulated apical views. Frequently, a modified four- and two-chambers views aiming posteriorly may be necessary to detect apical aneurysms and thrombus. Additionally, there is a broadly definition of aneurysm depending on criteria used, which may range from a small akinetic area to large with extensive wall thinning, similar to ischemic aneurysms18.Therefore, the low sensibility to detect aneurysm in FoCUS is expected"

The study should have included extra careful examination, using modified apical views aiming at more accurate diagnosis of apical aneurysms. Despite the small ones may not alter the LVEF, 3 they represent an abnormality related to the disease and an attentive observation is needed. 

Considering the "broad definition of aneurysm depending on the criteria used", it does not seem to me a proper argument, because one may always describe the alteration; additionally, the stored images were all subsequently analysed by experienced ecocardiographers (p. 5, 2nd paragraph, lines 14 and 15), who can standardise the definition of apical aneurysm.

Response: Although we agree that the use of extra careful examination using echocardiography modified apical views could improve the diagnostic accuracy of apical aneurisms, such evaluation is extremely difficult to perform with no technical specialist using focused eco study. As we pointed out in the discussion, to detect aneurysm several maneuvers for the appropriate assessment of apical region should be done during echocardiographic study. 

Regarding the argument related to the phrase "broad definition of aneurysm depending on the criteria used" we completely agree with you and this sentence has been removed from the Discussion section. 

4. P 12, 1st paragraph: "Right ventricular involvement is common in ChD and it is clear that there is a progression of RV dysfunction among patients with the various forms of ChD28. Nevertheless, the echocardiographic study of RV is challenging because of the anatomical and functional complexity of this chamber and the visual assessment of RV size and function could be difficult29... Our results demonstrated an excellent accuracy for RV size evaluation and a poor sensitivity..."

I would like to hear from the authors why they did not perform modified apical 4C view focused on RV: this would be easy, no time consuming, and would probably improve the sensitivity of both size and systolic function by means of eyeball evaluation.

Response: As previously discussed above, RV focused apical 4-chamber view is highly dependent on probe rotation which can result in an underestimation of RV width. Additionally, previous studies have been demonstrated the inaccuracy of eyeball RV function evaluation comparing with cardiac magnetic resonance. However, we agree with your comments and added this to the study limitation, on page 18, as follows: 

"Moreover RV focused apical 4-chamber view was not obtained, which would likely improve the sensitivity of both size and systolic RV function by means of eyeball evaluation.

5. FIGURE 2: Letters A, B, C, D, E and F are missing

Response: The letters have been added to Figure 2.

---

## [Editor Report · Decision Letter 1]

6 Oct 2021

Accuracy and reliability of focused echocardiography in patients with Chagas disease from endemic areas: SaMi-Trop Cohort Study

PONE-D-20-35550R1

Dear Dr. Perira Nunes,

We’re pleased to inform you that your manuscript has been judged scientifically suitable for publication and will be formally accepted for publication once it meets all outstanding technical requirements.

Kind regards,

Daniel A. Morris, M.D

Academic Editor

PLOS ONE

---

## [Editor Report · Acceptance letter]

25 Oct 2021

PONE-D-20-35550R1 

Accuracy and reliability of focused echocardiography in patients with Chagas disease from endemic areas: SaMi-Trop Cohort Study 

Dear Dr. Nunes:

I'm pleased to inform you that your manuscript has been deemed suitable for publication in PLOS ONE. Congratulations! Your manuscript is now with our production department. 

Kind regards, 

on behalf of

Dr. Daniel A. Morris 

Academic Editor

PLOS ONE